# 3D atomic structure from a single X-ray free electron laser pulse

Gábor Bortel ©[1] ✉, Miklós Tegze ©[1], Marcin Sikorski ©[2], Richard Bean ©[2], Johan Bielecki ©[2], Chan Kim[2], Jayanath C. P. Koliyadu ©[2], Faisal H. M. Koua ©[2], Marco Ramilli[2], Adam Round ©[2], Tokushi Sato ©[2], Dmitrii Zabelskii[2] & Gyula Faigel ©[1] ✉

X-ray Free Electron Lasers (XFEL) are cutting-edge pulsed x-ray sources, whose extraordinary pulse parameters promise to unlock unique applications. Several new methods have been developed at XFELs; however, no methods are known, which allow ab initio atomic level structure determination using only a single XFEL pulse. Here, we present experimental results, demonstrating the determination of the 3D atomic structure from data obtained during a single 25 fs XFEL pulse. Parallel measurement of hundreds of Bragg reflections was done by collecting Kossel line patterns of GaAs and GaP. To the best of our knowledge with these measurements, we reached the ultimate temporal limit of the x-ray structure solution possible today. These measurements open the way for obtaining crystalline structures during non-repeatable fast processes, such as structural transformations. For example, the atomic structure of matter at extremely non-ambient conditions or transient structures formed in irreversible physical, chemical, or biological processes may be captured in a single shot measurement during the transformation. It would also facilitate time resolved pump-probe structural studies making them significantly shorter than traditional serial crystallography.

XFEL sources were built to widen our knowledge of the matter at extremes. These sources promise the solution of structures of extremely small samples, even single molecules or small atom clusters[1], and the measurement of structural changes at extremely short timescales like 1–100 fs. In the last ten years, we have seen significant improvements in single particle imaging (SPI)[2,3]. However, the original expectations are not yet met. The resolution of these measurements is far from atomic. Another type of experiment for structural studies at XFEL-s is serial femtosecond crystallography (SFX)[4,5]. It is based on the same concepts as SPI: 1. measure before destruction and 2. serial measurements on thousands or millions of similar samples and combine the measured patterns to a single 3D intensity distribution. SFX is capable of producing datasets good for ab initio structure solution at the atomic level and it is widely used in biology, physics and chemistry.

SFX works better than SPI because it uses small crystallites with periodic structures and not an arbitrary arrangement of atoms like SPI. So far, we discussed the study of static structures. If we would like to go further and measure structures changing in very short time scales, we can use SPI and SFX but only in the pump-probe mode. This restricts these studies to processes which can be repeated millions of times exactly the same way. Therefore, we are mostly limited to laser pulse-driven processes. The ab initio solution of structures at extremely non-ambient conditions like very high magnetic field or very high pressure where one cannot repeat the same experiment exactly the same way is unreachable today. To solve the structure of matter in these cases one needs a method, which provides a dataset complete enough for ab initio structure solution from a single XFEL pulse. In this paper, we report a demonstration experiment that gives us the 3D atomic

[1]Wigner Research Centre for Physics, Institute for Solid State Physics and Optics, P.O.B. 49, 1525 Budapest, Hungary. [2]European XFEL GmbH, Holzkoppel 4, 22869 Schenefeld, Germany. ✉e-mail: bortel.gabor@wigner.hu; gf@szfki.hu

structure of crystalline materials from data taken during a single 25 fs XFEL pulse. To achieve this, we measured Kossel line patterns, collecting many Bragg reflections simultaneously. Although during the XFEL pulse the sample is destroyed similarly to SPI and SFX measurements, we can collect the Kossel pattern before the atomic structure changes. Due to interference with internal sources, in favorable cases, this method can give us not only the amplitudes of the structure factors but also their phases. This further simplifies the structure solution. In these demonstration experiments, we collected Kossel patterns of good-quality crystals of GaP and GaAs and could determine both the amplitudes and the phases of the structure factors. Therefore, the electron density was obtained with a direct Fourier synthesis. Although in these demonstration experiments, we used large, almost perfect single crystals, such measurements could be extended to non-perfect and small crystals. In these cases, we might lose the phase information, but we can turn back to the traditional evaluation methods of single crystal diffraction, where only the amplitude of the structure factor is used for the structure solution. Further developments in the experimental setup, especially on the detector side, would allow atomic resolution holography measurement[6–8]. This requires a very similar setup as the Kossel line measurement, and it would extend structural studies to systems with orientation order only.

## Result and discussion
### Kossel line pattern
Kossel lines are formed when atoms of a single crystal emit X-ray photons, and these are scattered by the crystal itself. This process is depicted schematically in Fig. 1 and Supplementary Movie 1. It is clear from the geometry and Bragg's law that the regions of modified intensity are conical sections on a flat 2D pixelated detector. Though the formation of Kossel lines was experimentally shown by Kossel, Loeck and Voges in 1935[9] and a theoretical description was given by

Laue in the same year[10], the method was not widely used in structural studies. The reason for this is that experimentally it is much more involved than traditional X-ray diffraction, it is limited to samples containing heavy atoms capable of emitting X-rays and the theoretical treatment of these patterns is complicated. Further, in normal circumstances (ambient conditions and long measuring times) it does not give more information than single-crystal X-ray diffraction. However, with the introduction of very intense pulsed sources (XFEL-s) and advanced 2D pixelated detectors, the simultaneous measurement of many Bragg reflections without the need of rotating the crystal, which

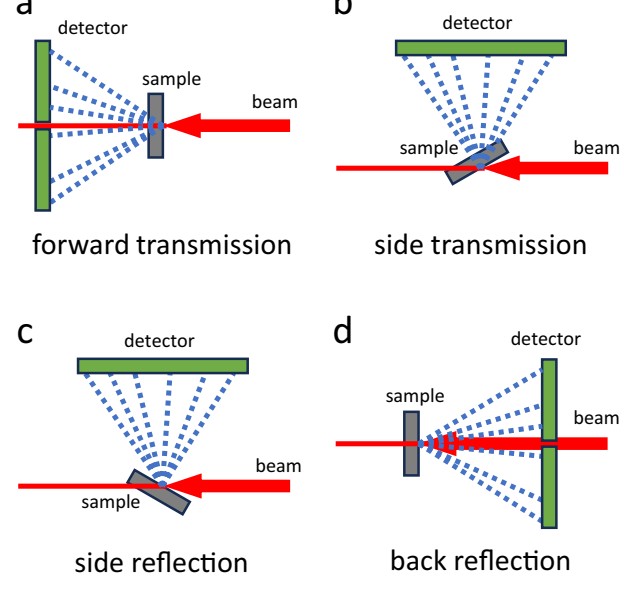

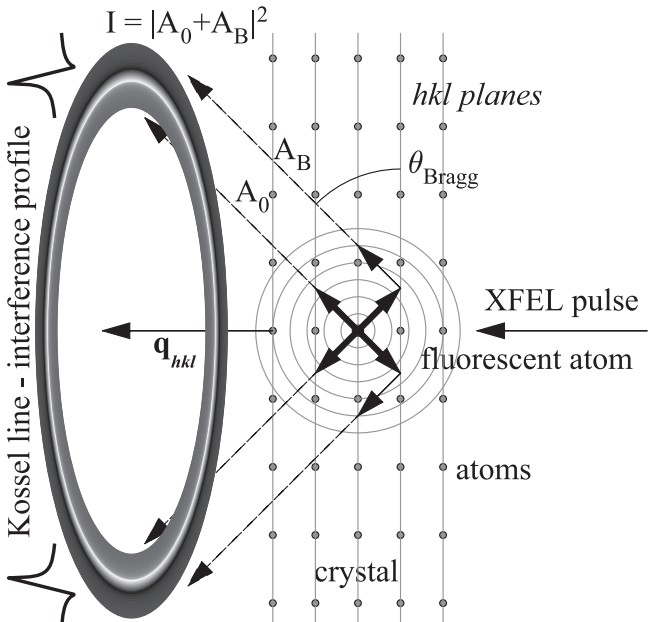

**Fig. 1 | Formation of Kossel lines.** Interference of X-rays emitted by atoms in a crystal and its Bragg-reflection encodes both the phase and amplitude of the structure factor in the intensity profile of the Kossel line. The incident x-ray beam excites the atoms in the sample, which in turn emit fluorescent photons as a spherical wave. Part of this wave is going out to the detectors directly, while an other part is scattered by the neighboring atoms and interferes with the direct beam. See also the animation of the illustrated processes in Supplementary Movie 1. The figure is adapted from our earlier publication[15].

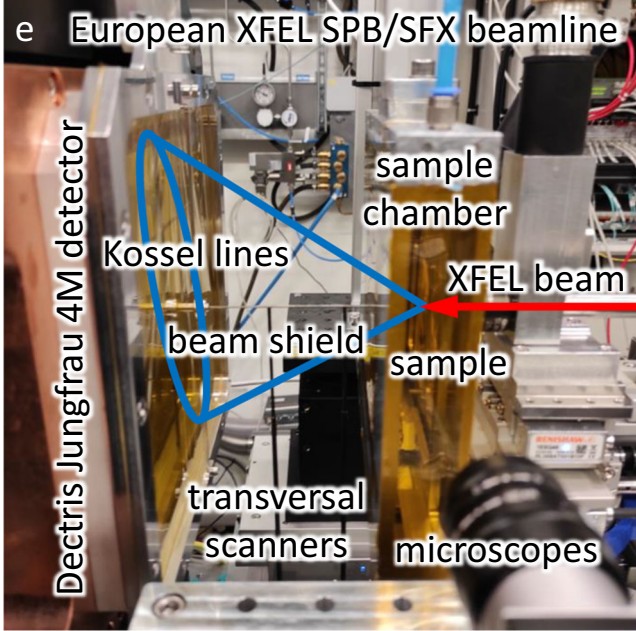

**Fig. 2 | Experimental arrangement. a, b, c, d** Various transmission and reflection geometries for recording Kossel line patterns from a single XFEL pulse. **e** The experimental setup in forward transmission geometry at the European XFEL. The red line indicates the collimated and focused X-ray beam that creates fluorescent radiation field leaving the sample. The blue ellipse schematically illustrates a Kossel line on the surface of the planar detector. The straight lines from the sample to the extremal points of the ellipse are two extremal generatrixes of a Kossel-cone, to emphasize the divergent, conical nature of the radiation field.

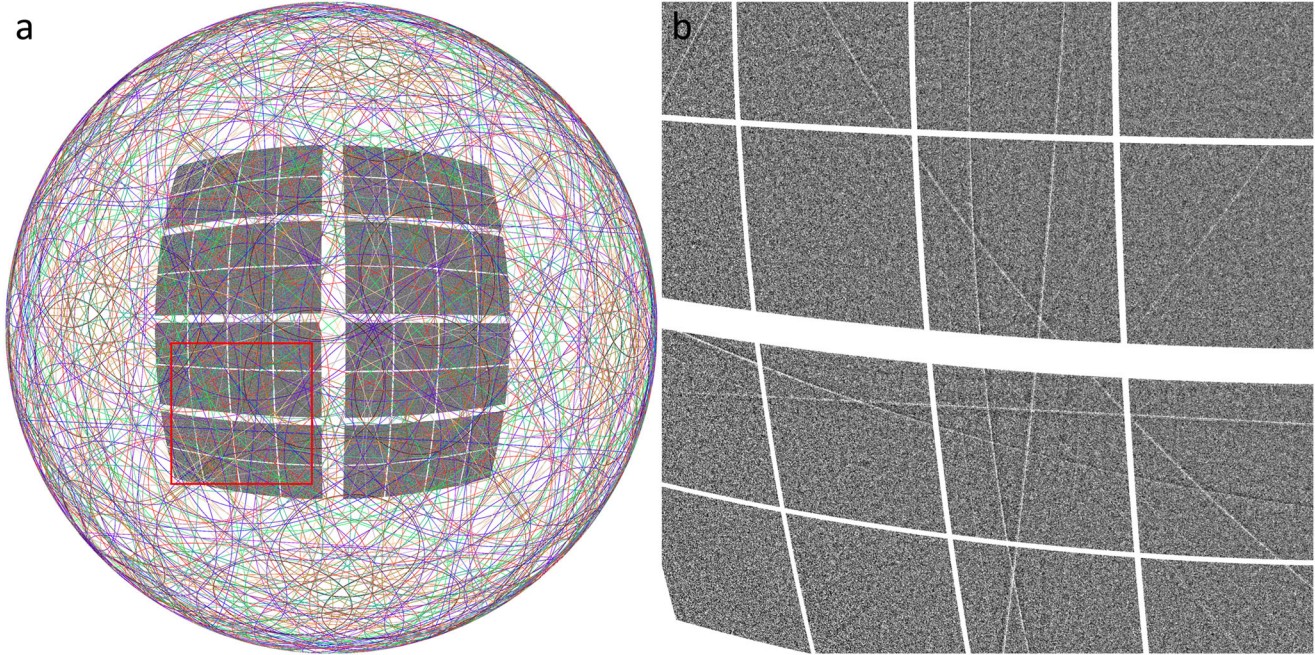

**Fig. 3 | Experimental Kossel line pattern. a** Complete Kossel line pattern and the observed fraction projected on a sphere. Lines of the same *d*-value have a common color. **b** Magnified region of the background-corrected normalized pattern (indicated by a red square on the complete pattern) reveals the stronger lines even to the naked eye. Source data are provided.

the Kossel technique provides, could lead to unique applications. One could collect a full diffraction data set in a very short time, in the extreme, during a single XFEL pulse. This facilitates the measurement of fast processes in single crystals. Since data collection is within a single pulse, we can think of ab initio determination of the changing structure during non-repeatable processes. Presently, this is not possible by any other method. Further, having good quality crystals, the fine structure of Kossel lines can provide the phases of the structure factors, eliminating the crystallographic phase ambiguity problem present in traditional crystallography. A theoretical description of the fine structure of Kossel lines for the x-ray Bragg case and in infinitely thick crystals has been developed by Hutton, Trammell and Hannon[11] within the framework of the dynamical theory of x-ray diffraction. However, for our purposes, this description is not enough, because we have finite thickness and many Laue case cones. Therefore, we turned to earlier works developed for nuclear sources and scatterers[12–14]. The theory worked out in these papers included the more general case of polarization mixing by the scattering medium and higher multipolarity sources. Since in our measurements, the sources are isotropic E1 emitters and the scattering is by electrons (*i.e.* electric dipole), we used a simplified form of the expressions given in these papers. For simplicity, we omitted polarization, which has negligible effects. Further, we applied the formulas for crystal slabs with finite thicknesses. We resorted to numerical solutions both in the Laue and in the Bragg case (See Supplementary Movie 2[15]). With these modifications, the fine structure of the Kossel lines could be analyzed; the amplitudes and phases for the measured reflections were obtained. A more detailed description of the Kossel lines and the formulas used in the evaluation process are given in the Supplementary Information file Supplementary Note 1. and their derivation in references[12–14].

## Experimental setup

We have developed the experimental procedure for x-ray holography for many years at synchrotron sources and could reach measuring times in the range of 1 s for a statistically meaningful pattern[8]. Kossel line pattern measurements require a very similar setup as inside source holography[8,15–17], except the spatial resolution of the 2D detector used for parallel detection of the fluorescent intensity forming the Kossel lines or the hologram. The short measuring time of holograms and Kossel lines at synchrotrons[18] prompted us to try the measurement of Kossel lines at XFEL sources. We realized the possibility of collecting a pattern during a single pulse. Starting from intensity considerations only, this conclusion seems trivial, since in the probe beam at a synchrotron we have about the same number of photons during 1 s as we have in a single XFEL pulse. However, the collection of all these photons in the very short time of an XFEL pulse with the precision necessary for distinguishing the lines from the background is not straightforward. The reason is that the detectors used at synchrotrons are counting detectors, while their XFEL counterparts are charge-integrating detectors. We have already experienced at synchrotrons that the detector is the weak point in the measurement[18]. Since at XFEL-s, the detector problem is even more pronounced, we tried to optimize all other experimental conditions for this demonstration experiment. First, we used samples (GaAs, GaP) from which we already collected good Kossel patterns or expected good-quality patterns. This allows us to check and strengthen the validity of XFEL measurements. Second, we choose the incident energy (10.5 keV) to excite only one element of the sample (Ga), increasing this way the signal to background ratio. Third, the sample was placed in He atmosphere in order to decrease air scattering. Unfortunately, we could not optimize two more parameters, the sample thickness and the experimental geometry. In the experiment, we used 100 micron thick samples in transmission geometry (Fig. 2 top left). None of the reflection geometries (Fig. 2 top right) were available due to technical limitations and the sample was thicker than the optimal 20–30 microns. Higher intensities obtained from thinner samples or in reflection geometry would cause saturation and malfunction of the detector. The implemented forward transmission arrangement (Fig. 2 bottom photo) yielded about 20/120 and 200/800 fluorescent photons/pixel/pulse at the edges/center of the 4 M Jungfrau detector[19] placed at 120 mm from the sample for GaAs and GaP, respectively. Due to the mismatch between the detector speed and the intra-train pulse repetition rate only a single pulse was used

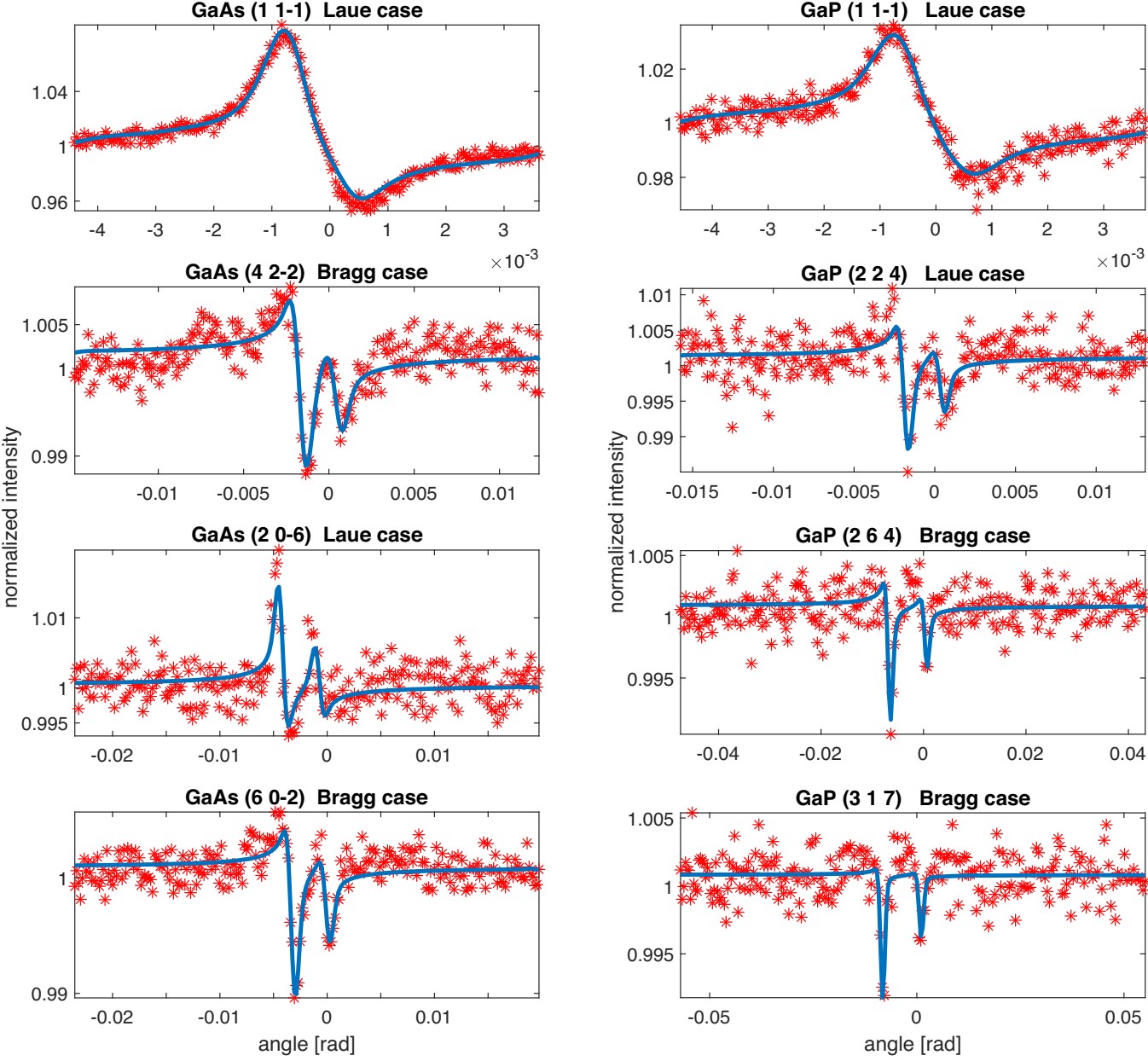

**Fig. 4 | Selected GaAs Kossel lines.** Line profiles of low (1 1 –1), medium (4 2 –2) and high (2 0 –6), (6 0 –2) index reflections for GaAs. Red stars indicate the profile data points extracted from the experimental Kossel line pattern. Blue lines indicate the fit of the theoretical model to the experimental profile. Source data are provided.

**Fig. 5 | Selected GaP Kossel lines.** Line profiles of low (1 1 –1), medium (2 2 4) and high (2 6 4), (3 1 7) index reflections for GaP. Red stars indicate the profile data points extracted from the experimental Kossel line pattern. Blue lines indicate the fit of the theoretical model to the experimental profile. Source data are provided.

from each train, namely data was acquired at 10 Hz. The spot size on the sample was set by compound refractive lenses to ~25 µm diameter. In one shot we had about 1 mJ total energy. Since pulses have different total energies because of the stochastic nature of the spontaneous emission, we took several shots, every shot at a new place of the sample to avoid the effect of radiation damage. With these beam parameters, we do not expect distortion of the Kossel lines caused by radiation damage. The radiation damage and possible nonlinear effects are discussed in more detail in the Supplementary Information file Supplementary Note 2. The sample motion was controlled by a fast x-y scanner, while the sample surface was checked by an optical microscope. Good shots were selected by visual inspection and statistical analysis of the recorded detector images later in the evaluation process.

## Analysis of Kossel line patterns

We have measured Kossel line patterns of GaAs and GaP single crystal samples using single 25 fs long XFEL x-ray pulses. By careful analysis of the Kossel line patterns, we could solve the structure of these samples ab initio. We arrived at the atomic structure in several steps: (i) background removal, (ii) Kossel line indexing, (iii) geometry refinement, (iv) profile extraction, (v) line profile fitting, (vi) Fourier transform of the complex structure factors. Details of these steps are described in the Supplementary Information file Supplementary Note 1. After background correction of the raw data, we identified and indexed the lines[20]. In Fig. 3, left panel, the indexed Kossel lines of GaAs produced by the Ga Kα$_{1,2}$ fluorescent photons are shown on a sphere together with the detector projected to this surface. Although lines caused

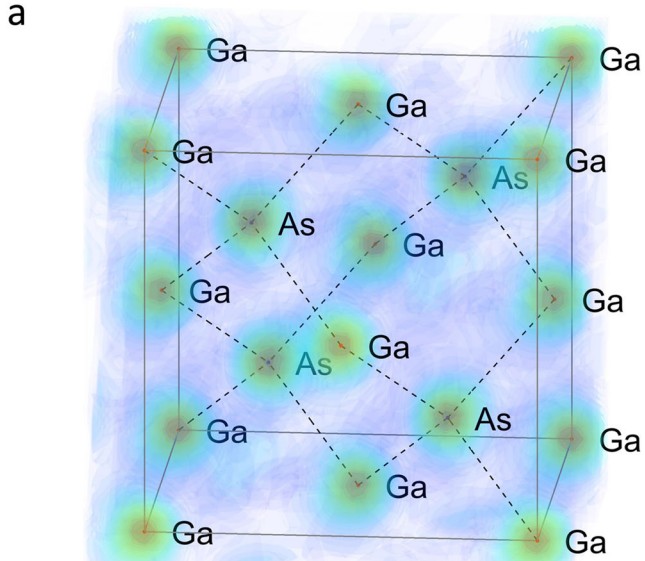

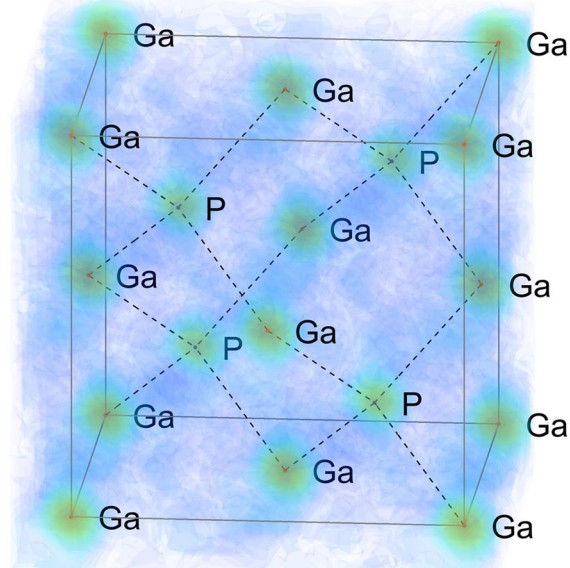

**Fig. 6 | Reconstructed electron densities. a** GaAs electron density obtained by direct Fourier synthesis from about 100 experimentally obtained structure factor amplitudes and phases. **b** Same for GaP, where the difference between Ga and P

electron density is also visible. For 3D rendering see Supplementary Movies 3 and 4. Source data are provided.

by the Kβ radiation are also visible on the image, we do not show them here for two reasons: (i) they are much weaker, and therefore their line shape could not be extracted with as good quality as that of the Kα₁,₂ lines (ii) we have not used them in obtaining the atomic structures. In Fig. 3, right panel an enlarged part of the measured pattern is shown without the theoretically calculated lines. Kossel lines can be clearly seen. Precise refinement of the Kossel line geometry, which is a prerequisite of the line-profile extraction yielded cubic lattice symmetry with lattice parameters of 5.655 Å and 5.453 Å for GaAs and GaP, respectively, in good agreement with literature values[21,22]. About 100 extracted line profiles had clearly identifiable, statistically meaningful fine structures (see a detailed inventory in the Supplementary Information file Supplementary Note 1.). These profiles were fitted with a theoretical Kossel line profile based on the dynamical theory of diffraction and amplitudes and phase angles of the reflections were obtained (see details in the Supplementary Information file Supplementary Note 1.). In Fig. 4, we show four typical line profiles of GaAs, two Bragg and two Laue cases for low, medium and high index reflections. Similarly, in Fig. 5, four line profiles for GaP are depicted. In the Fourier synthesis, we included all ~100 amplitudes and phases from the fitted profiles extending up to a cubic index of 8. All the other structure factors were taken as unknown and included with zero amplitudes. The assumed Friedel symmetry ensured the obtained electron density to be real. The forward scattering amplitude was taken as 184 for GaP and 256 for GaAs to define the zero level of the density.

Since we measured many shots on the same sample, we could double-check our results. We chose three shots from each sample, and we analyzed them independently. The exact number of fitted line profiles depended on the individual shot, because of the different statistics. The number of fitted profiles fluctuated between 85 and 120. In the reconstructed real space structure, the fluctuation in the number of used lines (the number of structure factors) did not cause any significant deviation. Figure 6 and the corresponding Supplementary Movies 3 and 4 show the reconstructed 3D real space structure of GaAs and GaP. Their structures are almost identical, except for the lattice spacing and the atom

types. In the case of GaAs, Ga, and As atoms could not be distinguished within the experimental error. This is not surprising since Ga has 31, while As has 33 electrons to scatter. Therefore, their scattering strengths are very similar. However, by improving the experimental conditions (using reflection geometry and detectors capable of handling more photons on the full surface of the detector), statistics will be much better. In future experiments, this will allow for extracting more precise structure factors leading to the possibility of distinguishing elements having atomic numbers close to each other. In the case of GaP, the two atom types have significantly different numbers of electrons, which reflects in the real space reconstruction, and the Ga and P atoms can be easily distinguished. To conclude this paragraph, we would like to point out that our demonstration measurement not only provides an opportunity to determine the atomic structure of phases formed during very fast, non-repeatable processes, but also lays the foundation for holographic measurements with atomic resolution[7]. It will widen the family of materials for single pulse structure determination to systems not having translation periodicity, only orientation order. So far, we mentioned those applications, which are unique to this method. However, there is a possibility to use our single pulse imaging technique in pump-probe experiments at XFEL-s. As it is clear from the description of the experiment, we could pump the sample by a laser beam before the X-ray pulse hits it, the same way as it is done in time resolved SFX experiments. However, in our case we do not have to measure tens of thousands of images of new samples but one pattern only (or a few taking into account the stochastic nature of the XFEL pulses and the less than 100% hit rate). This would significantly shorten the total measuring time, saving this way the very expensive XFEL beamtime. For more details on the prospects for possible applications, see the Supplementary Information file Supplementary Note 3.

In summary: the very intense and short XFEL pulses provide unique possibilities to deepen our understanding of various forms of matter. However, full exploitation of these possibilities requires special methods. We developed an experimental setup and the proper evaluation tools, which use a single very short and intense

XFEL pulse for ab initio structure determination. We demonstrated on GaAs and GaP single crystals that it is possible to solve their structure from data taken during a single 25 fs XFEL pulse. We recorded Kossel line patterns of these materials and used these for structure solution. Instead of a conventional crystallographic structure solution, we also demonstrated that the phases of structure factors can be extracted from the profile of Kossel lines, allowing direct Fourier synthesis of the electron density, and avoiding the phase ambiguity present in crystallography. This method might allow the study of crystalline phases formed during non-repeatable processes, facilitating measurements at extremely non-ambient conditions.

Further, it gives the bases to atomic resolution X-ray holographic measurements widening the types of materials suitable for single pulse structure determination. It could also make time-resolved pump-probe experiments on single crystals with sizes ~10 μm or larger at XFEL sources significantly shorter.

## Methods
The complete evaluation process from the raw detector image to the electron density of the solved structure is described and discussed in detail in the Supplementary Information file.

## Data availability
The raw and processed detector data, as well as other derived data generated in this study, are available in the Source Data file. The complete experimental data becomes publicly available at the European XFEL GmbH after an embargo period (SPB beamline, 202202/p003051 proposal). Source data are provided in this paper.

## Code availability
The procedures of the evaluation are described in the Supplementary Information file Supplementary Note 1. and in our previous publications[15,20]. The custom programming code used in the evaluation process is in development and unsuitable for public release. However, the programs used in this study are available from the corresponding author on request.

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

## Acknowledgements
We would like to thank B. Pécz and J. Volk for carrying out the thinning of GaAs and GaP wafers. Parts of this research were carried out at the European X-ray Free Electron Laser facility in Hamburg.

## Author contributions
All authors took part in the measurements at XFEL. In addition, G.F. came up with the idea of doing this measurement. He also did the theoretical work and the programming on the fine structure of the Kossel lines. He took part largely in the writing of the manuscript. G.B. developed the methods and software for online analysis, processing measured patterns, which allowed the removal of background, indexing the Kossel lines, refining the patterns and extracting the profile curves. He also took part in writing the manuscript and made all the illustrations. M.T. critically read the manuscript and discussed the theoretical work. G.F., G.B. and M.T. made the sample selection and preliminary preparation and characterization. M.S. and R.B. coordinated the work of the beamline personnel J.B., C.K., J.K., F.K., M.R., A.R., T.S., D.Z. who helped to run the beamline, and added their remarks to the manuscript.

## Funding

## Competing interests
The authors declare no competing interests.
