## [Peer Review File · Nature Communications]

3D atomic structure from a single X-ray free electron laser pulseREVIEWER COMMENTS

Reviewer #1 (Remarks to the Author):

In this work, G. Bortel and colleagues present a pilot experiment aimed at demonstrating the use of Kossel diffraction line patterns to resolve the 3D electron density in crystalline GaAs and GaP samples. The primary assertion of this study is that Kossel diffraction line analysis can serve as a viable alternative for three-dimensional reconstruction of electron density, addressing certain inherent challenges in time-resolved studies of single-particle imaging (which require multiple views of nearly identical samples) and/or serial femtosecond crystallography (which necessitates randomly oriented Bragg diffraction patterns). This concept holds potential interest, particularly in applications related to time-resolved studies of solid-state crystallographic samples, such as monitoring time-resolved liquid-to-solid phase transitions. However, in my personal opinion, I would recommend some possible improvements to the main text and SI:

(1) (1) Authors should comment on possible nonlinear effects on Kossel line intensity and phase modulation due to the high power deposited on the sample. For instance, in the early stages of femtosecond serial crystallography, A. Barty et al. (Nature Photonics 6 (1), 35-40, 2012) observed clear gating effects on the intensity of Bragg diffraction spots as a function of the energy deposited on the sample. I am curious about how this might impact Kossel analysis. Nonlinear effects could potentially influence the emission of fluorescence lines (for example, see S.K. Son et al., Physical Review Letters 107 (21), 218102, 2011) or could have detrimental effect if associated to directional effects in stimulated emission (for example, see J. Stöhr and A. Scherz Phys. Rev. Lett. 115, 107402 (2025)). Authors should provide more information on the FEL spot size and the density of energy deposited on the sample in the main text, this would be beneficial to understand the range of application of the methods. Additionally, it would be helpful to include information about the statistical spatial resolution of the reconstructed electron density for the reader.

(2) To better contextualize the article, authors should discuss how their approach could be applied to more complex scenarios. GaAs and GaP represent relatively simple zinc blende crystals. What are the limitations when dealing with unit cells that are more complex? Additionally, what if multiple fluorescence lines are present in the Kossel lines pattern, for instance, when the input photon energy exceeds the excitation energy for more than one species, especially in cases where the unit cell consists of more than two atoms? Would the detector be capable of resolving the different lines emitted by the sample, enabling the selection of individual species contributions and structural resolution?

(3) On the basis of this initial experiment, is it possible to extend Kossel analysis to biological crystals containing heavy atoms, or would the requirements for fluorescence emission and the weak scattering efficiency of C, N, and O atoms continue to limit the possibility of obtaining structural information even with the use of FEL radiation?

Minor comments:

(4) I guess that an explicit comment to ref 14 (Faigel, G., Bortel, G. & Tegze, M. Experimental phase determination of the structure factor from Kossel line profile. Scientific Reports 6, 22904 (2016). <https://doi.org/10.1038/srep22904>) in the caption of Figure 1 (as well in Figure S6 of SI) is necessary since these two schemes are identical to a previous publication of some of the authors.

(5) Same comment as (4) regarding the movie describing Kossel lines definition it is the same of movie 3 in SI of ref 14 (Faigel, G., Bortel, G. & Tegze, M. Experimental phase determination of the structure factor from Kossel line profile. Scientific Reports 6, 22904 (2016). <https://doi.org/10.1038/srep22904>).

(6) In Figure 2 I recommend the authors to increase the size of sketches on the top row.

(7) Would it be possible to calibrate the y-scale of Figures 4 and 5 (as well as in the SI for similar images) in terms of photons instead of normalized units? This adjustment would provide readers with a better understanding of the signal level in relation to the background.

(8) In the caption of Figure 6, I would use "3D rendering" instead of "3D information"

(9) In the captions of Figures S2 and S3, it is unclear what the authors mean by the following sentence: 'See high-resolution Supplementary Images of one detector pixel corresponding to one

image pixel, where the low and high values of the grayscale map are indicated in the filename.' Providing additional information for the reader may be necessary.

(10) Figure S4 contains an abundance of information, making it challenging to discern individual line profiles and convey the intended message to the reader. It might enhance clarity to vertically scale the different lines and/or present the GaAs and GaP cases as separate images.

(11) The data in Figure S5 are identical to those presented in Figures 4 and 5 of the main text. To avoid redundancy, it may be beneficial to either add 'reproduced from the main text' in the figure caption or refer solely to Figures 4 and 5 of the main text in the description of the fitting procedure.

(12) The description of the algorithm employed for fitting the Kossel profile on pages 7 and 8 bears significant resemblance to the corresponding section in reference 14 (Faigel, G., Bortel, G., & Tegze, M. 'Experimental phase determination of the structure factor from Kossel line profile.' Scientific Reports 6, 22904, 2016. <https://doi.org/10.1038/srep22904>). Although it's a technical aspect, using different wording for this section might be more appropriate.

Reviewer #2 (Remarks to the Author):

The authors have demonstrated the determination of 3D atomic structures from a single XFEL pulse by recording Kossel line patterns with a 2D detector and extracting the amplitude and phase of structure factors. To the best of my knowledge, this represents the first report of 3D atomic structure determination using a single-shot XFEL, and the novelty of the method is acknowledged. This achievement stands out, especially compared to conventional SFX, which relies on multiple XFEL shots to obtain 3D atomic structures. It is important to note, however, that the sample used in this demonstration, GaAs and GaP single crystals with known structures, renders the obtained structures themselves devoid of significant scientific implications.

The authors argue that this demonstrated method paves the way for investigating non-reproducible fast processes and structural transformations in crystals. Nevertheless, I have reservations regarding this claim for the following reasons:

Main Concerns:

[1] The ability of the proposed method to measure fast dynamics is constrained by the repetition frequency of XFEL (up to \sim MHz) or the detector's frame rate (likely \sim kHz with a 4 M Jungfrau detector). Any disparity between the XFEL pulse structure and the detector frame rate, as exemplified by the authors' demonstration at 10 Hz, would lead to even slower measurements. Consequently, measuring, e.g., sub-picosecond fast dynamics, as achieved in pump-probe experiments, appears unrealistic. If the authors maintain that fast dynamics can be measured, they should engage in a logical and quantitative discussion about the timescale of dynamical phenomena realistically measurable in the foreseeable future, taking the aforementioned constraints into account.

[2] The atomic structures derived from the Kossel line approach represent an average of neighboring structures across numerous fluorescence-emitter atoms within the sample. Therefore, the applicability of the proposed method for dynamic studies hinges on nearly all structures within the XFEL probe volume undergoing coherent and synchronous changes. While such dynamical phenomena are typically induced by pulsed lasers, as practiced in SPI and SFX pump-probe measurements, these processes often occur too fast for the proposed method. Should the authors consider other types of dynamical phenomena as examples, they should provide timescales supported by reference papers. In any case, these timescales should fall within the achievable temporal resolution mentioned in [1].

[3] In XFEL measurements, samples often suffer radiation damage upon exposure to a single XFEL shot, precluding the execution of dynamic measurements, as advocated by the authors, which necessitate multiple XFEL pulses at the same position within the sample. This point should be explicitly

stated in the paper. In conjunction with this, the authors should provide the XFEL focus size and intensity and assess the extent of sample damage following a single XFEL exposure in their demonstration experiment.

Other Comments:

[4] As elucidated in [2], the measurement of Kossel lines requires a substantial number of fluorescence-emitter atoms. Please specify the approximate number of fluorescent atoms required to obtain statistically significant Kossel line patterns and furnish details about the corresponding XFEL probe volume. While during approximately 1 second of X-ray exposure using a storage ring synchrotron radiation, a single fluorescent atom can be expected to undergo repeated excitation-deexcitation (fluorescence emission) cycles, this differs when employing XFELs with femtosecond pulse durations, necessitating consideration of the finite lifetime of the excited core-hole state. Such considerations would offer valuable insights into suitable XFEL focus sizes and intensities for these measurements.

[5] In lines 131-133, it is mentioned, "Since pulses have different energies because of the stochastic nature of spontaneous emission, we took several shots, every shot at a new place of the sample." This statement looks unreasonable. Why did the authors use new sample positions for each shot when XFEL pulse energies inherently fluctuate from pulse to pulse? This choice may have been necessitated by sample radiation damage, but clarification is warranted.

[6] Concerning the final atomic image presented in Figure 6, the authors should include a quantitative discussion on the precision/error associated with atomic positions, deviations from theoretical atomic image intensities, the presence of any artifacts in the images, etc. According to Figure S7, substantial errors exist in the determined phases, as large as 40 degrees.

[7] The text in the top row of images of Figure 2 appears excessively small, impeding readability. Likewise, the text in the bottom photo of Figure 2 is also difficult to discern.

[8] In the caption of Figure S3, "angles" should be removed from "structure factor amplitudes angles."

Reviewer #3 (Remarks to the Author):

The authors have successfully determined the 3D atomic structure of GaAs and GaP single crystals using data obtained from a single XFEL pulse. They developed an experimental setup and analysis tools for that. The results have never been achieved before, making them suitable for publication in Nature Communications. However, there are several points that must be addressed before the publication can proceed. Those are follows;

1. Radiation damage to the sample should occur during the exposure to the 25-fs XFEL pulse. Did the experimental data described here contain no such information? Please discuss the effect of radiation damage on their methods in the main text.

2. In the abstract, the authors described that their method can be applied in many ways, e.g., measuring transient structures formed in irreversible physical, chemical, or biological processes. But I think the range of applications is limited because the crystal sample must include heavy atoms. Please discuss the authors' prospects on that and the applicability of their method with specific examples in the main text.

3. In the analysis described here, the number of variables to be determined was sufficiently small compared to the amount of information contained in the experimental data, so the 3D atomic

structure determination using data obtained from a single XFEL pulse was probably successful. There may be difficulties in applying the method to more complex crystalline samples, so please discuss the authors' prospects for that in the main text.

4. The authors described that there was a difference of up to 40 degrees between the measured phases and the theoretical phases in Figure S7. Were those measured phases used as is in Fourier synthesis? Did they have any effect on the analysis results?

5. Please include a description of the dot color in the caption of Figure S8.

Answers to Reviewer #1

We thank the referee for his/her many important comments. We are sure that taking those into account will make the manuscript better.

Below we answer the referee's comments point by point.

In this work, G. Bortel and colleagues present a pilot experiment aimed at demonstrating the use of Kossel diffraction line patterns to resolve the 3D electron density in crystalline GaAs and GaP samples. The primary assertion of this study is that Kossel diffraction line analysis can serve as a viable alternative for three-dimensional reconstruction of electron density, addressing certain inherent challenges in time-resolved studies of single-particle imaging (which require multiple views of nearly identical samples) and/or serial femtosecond crystallography (which necessitates randomly oriented Bragg diffraction patterns). This concept holds potential interest, particularly in applications related to time-resolved studies of solid-state crystallographic samples, such as monitoring time-resolved liquid-to-solid phase transitions. However, in my personal opinion, I would recommend some possible improvements to the main text and SI:

(1) Authors should comment on possible nonlinear effects on Kossel line intensity and phase modulation due to the high power deposited on the sample. For instance, in the early stages of femtosecond serial crystallography, A. Barty et al. (Nature Photonics 6 (1), 35-40, 2012) observed clear gating effects on the intensity of Bragg diffraction spots as a function of the energy deposited on the sample. I am curious about how this might impact Kossel analysis. Nonlinear effects could potentially influence the emission of fluorescence lines (for example, see S.K. Son et al., Physical Review Letters 107 (21), 218102, 2011) or could have detrimental effect if associated to directional effects in stimulated emission (for example, see J. Stöhr and A. Scherz Phys. Rev. Lett. 115, 107402 (2025)). Authors should provide more information on the FEL spot size and the density of energy deposited on the sample in the main text, this would be beneficial to understand the range of application of the methods. Additionally, it would be helpful to include information about the statistical spatial resolution of the reconstructed electron density for the reader.

First, we comment on the XFEL spot size, since this is important in answering the first part of comment (1). We agree with the referee that we should have given explicitly the spot size in the main text. Therefore, we added 5 sentences to the Experimental section, and a reference [A. Barty et al., Nature Photonics 6, 35-40, 2012] with additional text relating our experiment to A. Barty's work in the Supplementary Information.

Here we answer the referee's comment in more detail. The spot size was 25 μm , which is about 10 times the spot size used in serial femtosecond crystallography measurements. Therefore, the deposited energy on unit area is 100 times smaller than in a typical serial crystallography experiment. In the paper mentioned by the referee (Barty et al.), the authors analyze their experimental findings, and they conclude that the diffraction pattern is not affected even at their much larger deposited energy, only the background increases. Since our measurement is also x-ray diffraction (although using inside sources) we do not expect deformation in the diffracted intensity pattern. Further, even the increase of the background is negligible because of the 100 times less deposited energy in unit area. The same is true for the nonlinear bleaching discussed in [S.K. Son et al., Physical Review Letters 107, 218102, 2011]. We had

about 100 times less photons/unit area as compared to what that paper uses to calculate changes in f'' . Therefore, the change in f'' is negligible; it will not appear in our experiment. As far as the third paper cited by the referee is concerned [J. Stöhr and A. Scherz, Phys. Rev. Lett. 115, 107402, 2015], we think it deals with a different energy range and different effect what we have in our experiment. This paper calculates the change of the Beer-Lambert law at XFEL experiments at low energies and for L edges of metals. They calculate the change in f'' close to the L3 resonance at about 700 eV and discuss how this affects the XMCD signal. This does not apply to our experiments, because we have less deposited energy than they calculated with, and more than 10 times higher photon energy, resulting about 1000 times smaller effect (because f'' scales with λ^3), Therefore this effect in our case is negligible. We think that this should not be discussed in the paper. We hope that the referee agrees with it. In the last sentence of point (1) the referee suggests including information about the statistical spatial resolution of the reconstructed electron density for the reader. We added more information about the spatial resolution in the Supplementary Information.

(2) To better contextualize the article, authors should discuss how their approach could be applied to more complex scenarios. GaAs and GaP represent relatively simple zinc blende crystals. What are the limitations when dealing with unit cells that are more complex? Additionally, what if multiple fluorescence lines are present in the Kossel lines pattern, for instance, when the input photon energy exceeds the excitation energy for more than one species, especially in cases where the unit cell consists of more than two atoms? Would the detector be capable of resolving the different lines emitted by the sample, enabling the selection of individual species contributions and structural resolution?

We answer (2) by going from simple to more complex systems.

(a), take a sample containing one heavy atom and many other atoms, which are not excited. In this case, the structure solution is going the same way as we did in the presented two cases. However, as the number of atoms increases in the unit cell one must measure more Kossel lines. As a rule of thumb, one should measure about 10 Kossel lines for one atom in the unit cell.

(b) We have a sample containing two different heavy atoms, which are excited. Although in principle it is possible to experimentally separate the emission lines if their energies are far enough, however we would not rely on this. Instead, we should sort out the Kossel lines corresponding to the emission energies by their different positions in the measured 2D image. This was already demonstrated in our earlier experiment performed at synchrotron on GaAs, where both the Ga and As emission lines were used. If the lines are sorted out this way, the solution of the structure is even more stable, because it works as if two independent measurements with different energies were done.

(c) There are two inequivalent sites of the same kind of atom in the unit cell and many other atoms, which are not excited. In this case, the phase of the structure factors cannot be determined without additional knowledge. We must know the relative position of the two excited atoms. In principle, this could be determined by fitting the relative position together with the phases for all lines in one iteration process. However, this would make the solution very complicated.

Connected to the applicability of the method, which is raised by the referee in this point there is one thing, which we did not mention in the manuscript because we concentrated on possibilities not reachable by other methods. Kossel pattern measurement would make XFEL pump-probe measurements simpler than using serial crystallography, since one does not have

to measure tens of thousands of images to get the structure, but one (or a few taking into account the not 100% hit rate). So, it would significantly shorten the time dependent studies in those cases where the samples are good for Kossel line formation. Shortening measurement time at XFEL-s is crucial, since XFEL beamtime is very expensive. Therefore, in the main text we added this application to the possible use of our method.

(3) On the basis of this initial experiment, is it possible to extend Kossel analysis to biological crystals containing heavy atoms, or would the requirements for fluorescence emission and the weak scattering efficiency of C, N, and O atoms continue to limit the possibility of obtaining structural information even with the use of FEL radiation?

We think that our approach is useable for biological crystals (mentioned implicitly in the introduction saying that we expect to be able to study biological processes). Light atoms should be seen. This is already shown in our GaP sample, where P atoms are much lighter than Ga but we could clearly see those. We estimate from the statistics of our present measurements that down to carbon we should see atoms. However, we do not expect to see hydrogen atoms. Further, we must note that atoms with close atomic numbers would not be easy to distinguish. In general, unit cells with many light atoms and only one heavy atom need better statistics. There is one more characteristic feature of biological crystalline samples, the not too good crystal quality. This would result in the loss of phase information. However, this does not limit structure solution more than the missing phase in traditional diffraction. In this case, one can use the well-established single crystal algorithms to solve the structure. Based on the remarks of the referee (point 2 and 3) we added a new section in the Supplementary Information about the limitation and possible application of the method.

Minor comments:

We agree with all minor comments (points 4-12) and did the suggested changes in the manuscript, at the same time taking into account the guidelines given by the journal.

(4) I guess that an explicit comment to ref 14 (Faigel, G., Bortel, G. & Tegze, M. Experimental phase determination of the structure factor from Kossel line profile. Scientific Reports 6, 22904 (2016). <https://doi.org/10.1038/srep22904>) in the caption of Figure 1 (as well in Figure S6 of SI) is necessary since these two schemes are identical to a previous publication of some of the authors.

We added the explicit comment and reference as the referee suggested.

(5) Same comment as (4) regarding the movie describing Kossel lines definition it is the same of movie 3 in SI of ref 14 (Faigel, G., Bortel, G. & Tegze, M. Experimental phase determination of the structure factor from Kossel line profile. Scientific Reports 6, 22904 (2016). <https://doi.org/10.1038/srep22904>).

We added the explicit comment and reference as the referee suggested.

(6) In Figure 2 I recommend the authors to increase the size of sketches on the top row.

We rearranged the top part of Figure 2 and doubled the size of sketches.

(7) Would it be possible to calibrate the y-scale of Figures 4 and 5 (as well as in the SI for similar images) in terms of photons instead of normalized units? This adjustment would provide readers with a better understanding of the signal level in relation to the background.

The use of the normalized units when working with Kossel line profiles is twofold. (1) Each point in a Kossel line profile has contributions from different regions of the raw detector image with largely different photon counts (due to air scattering, absorption and geometry, as it is explained in the manuscript and shown in Figure 1). (2) Each point in a Kossel line profile has a different number of contributing detector pixels (depending on its location within the solid angle covered by the detector, as shown in the bottom panels of Figure S4). Therefore, the overall profile cannot be derived in phonon count units.

Still, an „average“ photon count can be estimated for the Kossel line profiles as follows: The raw images have ~80 and ~500 photons/pixel on average for GaAs and GaP, respectively. In both cases the extracted profiles have typically 300 contributing detector pixels for a Kossel line profile point. These result in 24000 and 150000 photons contributing to a Kossel line profile point on „average“. It is worth to consider the Poisson statistics of these typical photon counts. Their relative uncertainty ($1/\sqrt{N}$) is 0.0065 and 0.0026, for GaAs and GaP respectively. These are well below the typically few percent amplitude of the extracted Kossel line amplitudes, indicating that our result is statistically meaningful.

This estimate was added to the „Profile extraction“ paragraph of the Supplementary Information.

(8) In the caption of Figure 6, I would use “3D rendering” instead of “3D information”

We changed the wording as suggested by the referee.

(9) In the captions of Figures S2 and S3, it is unclear what the authors mean by the following sentence: 'See high-resolution Supplementary Images of one detector pixel corresponding to one image pixel, where the low and high values of the grayscale map are indicated in the filename.' Providing additional information for the reader may be necessary.

In the captions of Figure S1 and S2 we refer to the full resolution lossless Supplementary Images. Their size (2178×2170 pixels) exactly corresponds to the number of detector pixels. Low and high values of the colormap/grayscale map corresponding to photons/pixel for Figure S1 and normalized units for Figure S2 are indicated in the filenames, as these should not be put on the images. These images are provided to let the interested reader closely

investigate the Kossel line patterns, or even use as raw experimental data. We hope that the modified figure captions are clearer now.

(10) Figure S4 contains an abundance of information, making it challenging to discern individual line profiles and convey the intended message to the reader. It might enhance clarity to vertically scale the different lines and/or present the GaAs and GaP cases as separate images.

The purpose of this figure is just to illustrate the profile-extraction step of the data evaluation chain, not to show the individual profiles. It shows how the profiles look, if transformed on a common x axis by transforming the θ Kossel cone half opening angles to $2d\sin\theta = \lambda$, the emission wavelength. Also, the bottom panel shows how the statistics deteriorate when the number of contributing pixels drops to zero. This figure is actually a screenshot of our program, where the shown lines can be filtered based on various attributes. To make the figure less busy, only the lines up to 1.4\AA resolution are shown.

(11) The data in Figure S5 are identical to those presented in Figures 4 and 5 of the main text. To avoid redundancy, it may be beneficial to either add 'reproduced from the main text' in the figure caption or refer solely to Figures 4 and 5 of the main text in the description of the fitting procedure.

To have the steps of the data evaluation process in one document, we choose the first option the referee suggested and kept the image repeated in the Supplementary Information and added a remark to the caption of Figure S5.

(12) The description of the algorithm employed for fitting the Kossel profile on pages 7 and 8 bears significant resemblance to the corresponding section in reference 14 (Faigel, G., Bortel, G., & Tegze, M. 'Experimental phase determination of the structure factor from Kossel line profile.' Scientific Reports 6, 22904, 2016. <https://doi.org/10.1038/srep22904>). Although it's a technical aspect, using different wording for this section might be more appropriate.

We agree that in the explanation of the formulas we used very similar wording as in the publication mentioned by the referee. The reason for this is that we used the same theoretical framework for evaluation as in our earlier paper. We made only slight changes, which are mentioned in the present description. We could leave this section fully out and cite our earlier publication. However, we wanted to make it simpler for the readers to go through our reasoning without searching references. However, we made changes in the text where it was possible to satisfy the referee's suggestion.

Answers to Reviewer #2

We thank the referee for the very helpful comments, especially the first comment, which prompted us to mention another possible application we did not anticipate in the original text.

Below we answer the referee's comments point by point.

The authors have demonstrated the determination of 3D atomic structures from a single XFEL pulse by recording Kossel line patterns with a 2D detector and extracting the amplitude and phase of structure factors. To the best of my knowledge, this represents the first report of 3D atomic structure determination using a single-shot XFEL, and the novelty of the method is acknowledged. This achievement stands out, especially compared to conventional SFX, which relies on multiple XFEL shots to obtain 3D atomic structures. It is important to note, however, that the sample used in this demonstration, GaAs and GaP single crystals with known structures, renders the obtained structures themselves devoid of significant scientific implications.

The authors argue that this demonstrated method paves the way for investigating non-reproducible fast processes and structural transformations in crystals. Nevertheless, I have reservations regarding this claim for the following reasons:

Main Concerns:

[1] The ability of the proposed method to measure fast dynamics is constrained by the repetition frequency of XFEL (up to ~ MHz) or the detector's frame rate (likely ~ kHz with a 4 M Jungfrau detector). Any disparity between the XFEL pulse structure and the detector frame rate, as exemplified by the authors' demonstration at 10 Hz, would lead to even slower measurements. Consequently, measuring, e.g., sub-picosecond fast dynamics, as achieved in pump-probe experiments, appears unrealistic. If the authors maintain that fast dynamics can be measured, they should engage in a logical and quantitative discussion about the timescale of dynamical phenomena realistically measurable in the foreseeable future, taking the aforementioned constraints into account.

If one wants to use the XFEL pulses to induce some change in the sample and wants to study this change, then the referee is right, that the pulse structure is the limiting factor of the time resolution. However, we never mentioned this possibility.

In the introduction of the manuscript, we write:

„This type of measurement opens the way for studying non-repeatable fast processes and structural transformations in crystals for example measuring the atomic structure of matter at extremely non-ambient conditions or transient structures formed in irreversible physical, chemical, or biological processes.”

In the conclusion, we write:

„This method opens new avenues in atomic-level structural studies. It might allow the study of crystalline phases formed during non-repeatable processes, facilitating measurements at extremely non-ambient conditions. Further, it gives the bases to atomic resolution x-ray holographic measurements widening the types of materials suitable for single pulse structure determination.”

We emphasized that one could be able to determine the 3D structure of matter in such cases when no repeatable pump-probe measurements could be done. We also defined what we mean

by this: extremely non-ambient conditions. To be clear, we give two examples here:
(a) Many studies aim to find the atomic structure of matter at very high pressures (for example pressures present at the interior of planets Earth, Jupiter etc.), which is difficult to statically maintain. In these cases, we cannot do pump-probe experiments in easily repeatable and well-defined way. However, we can make this large pressure for a very short time (ms-ns). However, there is no method which could determine the atomic structure at these cases from one single measurement. Our approach could do it.

(b) Other example is very high magnetic fields. If one intends to study matter in very high magnetic fields, which cannot be produced as a static field, but can be produced as one very short pulse, again there is no method, which could determine the 3D atomic arrangement induced by this field.

We think that both the introduction and the conclusion part is clear on this point. At last, we call the attention of the referee to a collection of articles by Nature about high pressure works [10. November 2022] and a report on high magnetic field research in the US [ISBN 978-0-309-38778-1 | DOI 10.17226/18355]. Both discuss studies from solid state physics through biology to planetary science involving high pressure and high magnetic field research including pulsed methods.

We believe that our method could significantly widen the possibilities for structural studies on these fields.

One more remark to this point: the referee mentioned the pump-probe experiments. Using our measurement in XFEL pump-probe experiments, we can reach the same time resolution as done in serial crystallography. However, we do not have to take thousands or millions of diffraction patterns at a given time point but one (or a few if we take into account the stochastic nature of the beam intensity, and good sample position for the hit). That makes these time-dependent studies much faster and much easier using our approach. The price we pay for it compared to serial crystallography is the sample size. We need samples in the 10 μm range not in the submicron regime. Therefore, our method will not replace serial crystallography for small samples but could be a good choice for larger samples. We added this type of application to the ones we have already mentioned in the main text. We note here that we used 10 Hz because for the demonstration experiment, we did not need many pulses close to each other; we intended to show that atomic resolution 3D structure can be determined from a single pulse. Further, the choice of JUNGFRU detector does not mean that we cannot use for example the AGIPD, which has higher frame rate and is mostly used in serial crystallography measurements. The 4M AGIPD, which according to our knowledge will be available from the beginning of 2024 perfectly fits our measurement. Therefore, we can work with the same time resolution as any other measurement done at XFEL-s.

To give a clearer view for the readers about the possible applications we added a paragraph to the Supplementary Information. There we explain in more detail what type of samples could be studied, what are the limitations of the method and what would be the most trivial applications.

[2] The atomic structures derived from the Kossel line approach represent an average of neighboring structures across numerous fluorescence-emitter atoms within the sample. Therefore, the applicability of the proposed method for dynamic studies hinges on nearly all structures, within the XFEL probe volume undergoing coherent and synchronous changes. While such dynamical phenomena are typically induced by pulsed lasers, as practiced in SPI and SFX pump-probe measurements, these processes often occur too fast for the proposed method. Should the authors consider other types of dynamical phenomena as examples, they

should provide timescales supported by reference papers. In any case, these timescales should fall within the achievable temporal resolution mentioned in [1].

The first part of the remark states that what we measure is an average of the illuminated volume. That is absolutely true. In every method used at XFEL diffraction studies what we measure is an average of the illuminated volume. So, this is the same for SPI and SFX as mentioned by the referee. Further, the referee writes: „While such dynamical phenomena are typically induced by pulsed lasers, as practiced in SPI and SFX pump-probe measurements, these processes often occur too fast for the proposed method.” We do not understand this sentence. Our method is as fast as the XFEL pulse allows. We explained in connection with the previous remark (1) that we can do the same pump probe experiments as done by SFX. If a process is measurable by SFX it is measurable by our approach, with the advantages mentioned in point (1). However, we never mentioned in the manuscript that we want to compete with SFX in those cases, which can be done by SFX. We pointed out that our approach allows measurements, which was not possible so far at all. We explained this in point (1), and it is also formulated in the introduction, conclusion and in the part of the Supplementary Information, which we added as mentioned in point (1). Therefore, we think that we do not have any limiting factor in time resolution as compared to other methods used for time resolved studies presently.

[3] In XFEL measurements, samples often suffer radiation damage upon exposure to a single XFEL shot, precluding the execution of dynamic measurements, as advocated by the authors, which necessitate multiple XFEL pulses at the same position within the sample. This point should be explicitly stated in the paper. In conjunction with this, the authors should provide the XFEL focus size and intensity and assess the extent of sample damage following a single XFEL exposure in their demonstration experiment.

The referee is right about the radiation damage. It is present in all measurements at XFEL-s. It prevents dynamical studies of the same sample at the same spot. That is true for all measurements. That is why the samples are refreshed in SPI and serial crystallography measurements. However, it is also true, that one can obtain information on the original structure of the sample if the XFEL pulses are short enough. In that case, one collects the elastically scattered photons before the atoms have time to move. The first paper suggesting this type of measurement was published in 2000 [see ref. 1 in main text]. In the following years many papers were published studying the time evolution of atomic motions caused by an XFEL pulse [see for example ref. 9 in Supplementary Information]. Since the time scale of the fluorescent process is below 1 fs, the limiting time for our measurement is the length of the XFEL pulse, which is much longer (in our case 25 fs) than the characteristic time of the pattern forming radiation. Therefore, the radiation damage affects our measurement the same way as it affects any SPI or SFX measurement. It is shown by many SFX experiments that radiation damage does not prevent atomic resolution structure determination. Therefore, we do not expect any problem caused by radiation damage.

To explain the effect of radiation damage to the reader, we added 5 sentences about radiation damage in the main text at the end of the Experimental section. There we also gave the focal spot size (25 μm diameter). Further, we added a section about radiation damage in the Supplementary Information.

Other Comments:

[4] As elucidated in [2], the measurement of Kossel lines requires a substantial number of fluorescence-emitter atoms. Please specify the approximate number of fluorescent atoms required to obtain statistically significant Kossel line patterns and furnish details about the corresponding XFEL probe volume. While during approximately 1 second of X-ray exposure using a storage ring synchrotron radiation, a single fluorescent atom can be expected to undergo repeated excitation-deexcitation (fluorescence emission) cycles, this differs when employing XFELs with femtosecond pulse durations, necessitating consideration of the finite lifetime of the excited core-hole state. Such considerations would offer valuable insights into suitable XFEL focus sizes and intensities for these measurements.

In the incident beam we had $\sim 10^{12}$ photons/pulse. Since our sample was 100 μm thick, almost all the photons were absorbed by the sample. We can take 100% absorption resulting in a slight overestimation. The number of atoms in the illuminated area (25 μm spot size) was approximately 10^{16} . This means that on average 1 of 10 000 atoms will be excited in the sample. The probability of higher excitation is very low at these beam parameters. Therefore, we do not expect finite lifetime effects in our experiments.

[5] In lines 131-133, it is mentioned, "Since pulses have different energies because of the stochastic nature of spontaneous emission, we took several shots, every shot at a new place of the sample." This statement looks unreasonable. Why did the authors use new sample positions for each shot when XFEL pulse energies inherently fluctuate from pulse to pulse? This choice may have been necessitated by sample radiation damage, but clarification is warranted.

We took several shots, because different shots have different number of incident photons resulting in images with different statistics. In some images one could not see the Kossel lines with good enough statistics, to determine their shapes and positions precisely enough. We wrote this in the manuscript: lines 134-136. „Good shots were selected by visual inspection and statistical analysis of the recorded detector images later in the evaluation process.” We measured at different places, because the beam destroys the sample, basically it makes a hole. However, this process is much slower than the 25 fs length of the XFEL pulse. To clarify this, we added 5 sentences at to the end of the Experimental section, and discussed the radiation damage effects in a new section of the Supplementary Information.

[6] Concerning the final atomic image presented in Figure 6, the authors should include a quantitative discussion on the precision/error associated with atomic positions, deviations from theoretical atomic image intensities, the presence of any artifacts in the images, etc. According to Figure S7, substantial errors exist in the determined phases, as large as 40 degrees.

We added a calculation, a new figure and discussion on the precision of atomic positions, and electron density values in the Supplementary Information (Figure S10).

[7] The text in the top row of images of Figure 2 appears excessively small, impeding readability. Likewise, the text in the bottom photo of Figure 2 is also difficult to discern.

We doubled the size and rearranged the top part of Figure 2 according to the suggestion of the referee.

[8] In the caption of Figure S3, "angles" should be removed from "structure factor amplitudes angles."

We removed the „angles”, as the referee suggested.

Answers to Reviewer #3

We thank the referee for the very useful comments. We use them to improve the manuscript.

Below we answer the referee's comments point by point.

The authors have successfully determined the 3D atomic structure of GaAs and GaP single crystals using data obtained from a single XFEL pulse. They developed an experimental setup and analysis tools for that. The results have never been achieved before, making them suitable for publication in Nature Communications. However, there are several points that must be addressed before the publication can proceed. Those are follows;

1. Radiation damage to the sample should occur during the exposure to the 25-fs XFEL pulse. Did the experimental data described here contain no such information? Please discuss the effect of radiation damage on their methods in the main text.

We estimated the possible radiation damage before the experiment by comparing the experimental parameters to previous XFEL experiments, and to theoretical works on radiation damage. We found that the radiation damage has no effect on the the shape of the Kossel lines. The effect of the x-ray pulse develops much later than the collection of the elastically scattered photons (which is ~25 fs, the length of the pulse). To explain this to the reader we added and changed 5 sentences in the main text at the end of the Experimental section and a paragraph in the Supplementary Information discussing the radiation damage was also added.

2. In the abstract, the authors described that their method can be applied in many ways, e.g., measuring transient structures formed in irreversible physical, chemical, or biological processes. But I think the range of applications is limited because the crystal sample must include heavy atoms. Please discuss the authors' prospects on that and the applicability of their method with specific examples in the main text.

The referee is right that the applications are limited because the crystalline samples should contain heavy atoms. Since the other referees also asked about possible applications, we added a detailed discussion about limitations and possibilities in the Supplementary Information. We also changed the text dealing with applications in the main text (introduction and conclusion).

We hope that the referee agrees that the detailed discussion of possible application fits better into the Supplementary Information.

3. In the analysis described here, the number of variables to be determined was sufficiently small compared to the amount of information contained in the experimental data, so the 3D atomic structure determination using data obtained from a single XFEL pulse was probably successful. There may be difficulties in applying the method to more complex crystalline samples, so please discuss the authors' prospects for that in the main text.

Yes, the referee is right that we had many more Kossel lines (structure factors) than the minimum, necessary for the simple structures we determined. We agree with the referee that it is more difficult to measure more complex crystalline samples. We discuss the limitation through examples in more detail in the Supplementary Information, in a separate paragraph titled: „Prospects for possible applications”. We think that this longer discussion does not fit in the main text, since our manuscript reports an experiment, the main message is that one can measure a 3D structure in 25 fs using an XFEL source. We hope that the referee agrees that the detailed discussion of possible application fits better into the Supplementary Information.

4. The authors described that there was a difference of up to 40 degrees between the measured phases and the theoretical phases in Figure S7. Were those measured phases used as is in Fourier synthesis? Did they have any effect on the analysis results?

Indeed, some of the structure factors had 40 degrees deviation as compared to the theoretical phases. We used all the measured structure factors, where we could evaluate the phase and magnitude of the structure factors including those having large deviations. Of course, these structure factors contribute to the overall solution. However, the number of structure factors with this high error are small. Most of the structure factors have less than 20 degrees deviation as compared to the theoretical values. This precision is enough to obtain a correct atomic structure. Since the weight of the structure factors with larger errors is small in the Fourier synthesis, we do not expect large effect caused by them. The deviation of the phases and the amplitudes of the structure factors compared to the theoretical values is reflected in the intensity found between atomic positions, as seen in Figure 6. This is normal in all single crystal measurements. However, as it is clear from the figure, the atoms are well out from the noise.

5. Please include a description of the dot color in the caption of Figure S8.

We changed the caption according to referee's comment.

REVIEWERS' COMMENTS

Reviewer #1 (Remarks to the Author):

The authors have replied to all my questions and they largely improve the manuscript in according to the inputs of all three referees. I have no additional comments or request of explanation. I recommend the paper for publication.

Reviewer #2 (Remarks to the Author):

By reading the authors' responses to my main concerns, I understand that my intent was not correctly conveyed to them. For instance, in the Conclusions section, the sentence "It might allow the study of crystalline phases formed during non-repeatable processes" could be interpreted by many readers as obtaining different atomic structures at various time delays during single non-repeatable processes. However, the authors' responses indicate they do not consider providing multiple atomic structures for a single non-repeatable process. The manuscript is especially misleading as it combines discussions about pump-probe measurements for recording molecular movies with measurements implicitly limited to a single atomic structure snapshot. To avoid any misunderstandings among readers, it is essential to explicitly state in the Abstract, Introduction, and Conclusion of the main text, where non-repeatable processes are mentioned, that the sample is destroyed by a single XFEL shot, and thus, only one atomic structure snapshot can be obtained per non-repeatable process. This clarification is necessary for the publication of the manuscript.

Reviewer #3 (Remarks to the Author):

The authors have added discussions of my concerns about radiation damage and about applicability of their method in the Supplementary Information. Long discussions on applicability/limitation of their method do not need to be in the main text, but it is better to clearly state in the main text that such discussions are in the Supplementary Information. If that is the case, I have no further comments and I think this paper is worth publishing in Nature Communications.